# Asymmetric modeling of regional tourism economic disparity in China

**Huan Ling** [1]*, **Xiaoyue Qiu**[2]

**1** Tourism College, Sichuan University, Chengdu, China, **2** School of Economics and Management, Southeast University, Nanjing, China

* linghuan42@sina.com

## Abstract

China's tourism economy has experienced significant regional disparities. However, little attention has been paid to understanding the intricate mechanisms through which the interplay of influential factors gives rise to such disparities. Utilizing statistical data from the tourism economy of 31 provinces in mainland China, this study investigates regional tourism economic disparity through the asymmetric modeling of fuzzy set qualitative comparative analysis (fsQCA). The findings reveal that the causes of regional tourism economic disparity exhibit asymmetry; varying approaches contribute to the formation of high and low levels of tourism economy; the functioning of the most influential factors is impacted by other antecedent conditions, presenting an asymmetric non-linear effect; favorable transportation convenience is a necessary condition for a high level of tourism economy, while poor tourism attraction is a necessary condition for a low level of tourism economy. This research not only introduces a fresh perspective on regional tourism economic disparities, enabling an in-depth comprehension of the complex interactions and nonlinear functional mechanisms of influential factors, but also explores a region-based tourism development model, offering valuable practical guidance for policymakers in the tourism sector.

## Introduction

As a thriving industry, tourism serves as a major economic driver in many countries [1]. Since the implementation of the reform and opening-up policy in 1978, China's tourism has gradually integrated into the national economy, evolving into a strategic pillar industry [2, 3]. According to the Ministry of Culture and Tourism of the People's Republic of China, tourism contributed 10.94 trillion Yuan to the GDP, accounting for 11.05% of the total GDP, while direct and indirect employment in tourism comprised 10.31% of China's total employment in 2019. Internationally, China has one of the largest domestic, inbound, and outbound tourism markets [4], ranking first as a source of outbound tourists and fourth as a receiving country for inbound tourists over many consecutive years. Despite these achievements, China's tourism grapples with uneven development, exemplified by economic disparity between its inland and coastal regions [5, 6]. This disparity objectively reflects unbalanced and inadequate development in China, a principal contradiction that the country endeavors to address. Examining

and China Economic and Social Big Data Research Platform (https://data.cnki.net/).

**Funding:** This study was supported by the Soft Science Research Project of Chengdu, Sichuan Province, China (Grant No. 2020-RK00-00361-ZF). The funders had no role in the study design, data collection and analysis, decision to publish, or preparation of the manuscript.

**Competing interests:** The authors have declared that no competing interests exist.

the formation mechanism of China's regional tourism economic disparity has played a pivotal role in narrowing the regional tourism economic gaps and fostering balanced regional development [7–9]. Although this study primarily focuses on China's regional tourism economic disparity, a common phenomenon in global tourism development [10–12]—it can still capture the interests of other countries, particularly developing nations.

Numerous studies have been conducted on China's regional economic disparities in tourism [6, 13]. For instance, Wang et al. [12], Zhang et al. [14], and Zhu et al. [15] empirically validated this disparity by analyzing its spatial features and influential factors. These studies have contributed significantly to our understanding of economic disparities in China's regional tourism. However, previous studies have certain limitations. First, most studies have focused solely on measuring the temporal and spatial differences in the regional tourism economy, clearly indicating its temporal and spatial distribution and alterations, but have failed to explore the causes of these differences [12]. Second, inbound tourism development is often the primary research focus, with China's domestic tourism rarely included [5, 16, 17]. However, in reality, contributing more than 82.80% of China's total tourism revenue over the past five years, the domestic tourism market, rather than the inbound market, is the most crucial aspect of China's tourism development [7]. Third, while some researchers have investigated the factors influencing regional tourism economic disparity [13, 18], an in-depth analysis of their interactions and interrelationships is lacking. Regression analysis, which is often based on mutual independence among independent variables, one-way linear relationships, and symmetric causality, is commonly used to verify causality. However, this approach makes it difficult to reveal multiconcurrent causality, asymmetric causality, and other complex causality types incorporated into the interaction of influential factors that contribute to economic disparity in regional tourism [13, 19].

To address these limitations, this study introduces a fuzzy set qualitative comparative analysis (fsQCA), which is suitable for handling asymmetric causality. It incorporates the influential factors of regional tourism economic disparity into an asymmetric model, revealing the complex interactions and nonlinear relationships among them. This approach breaks away from the constraints of the traditional symmetric causality analysis, systematically clarifying its effects and explicating the complex causality of multiple factors that jointly influence regional tourism economic disparity. Furthermore, this study considers both domestic and inbound tourism to comprehensively reflect the economic disparity in China's regional tourism.

Considering that provinces are fundamental units in China's regional context, and approaches based on provinces are commonly employed in research on China's regional tourism economic disparity [5, 6], this study utilized 31 provinces in mainland China as the analysis samples. They extracted tourism-related statistical data from 2005 to 2019. Due to the significant impact of the COVID-19 pandemic, tourism industry data from the past three years are considered abnormal. This study examines the regional tourism economic disparity and causality. Theoretically, this study constructs an exploratory asymmetric causality model to analyze regional tourism economic disparity and proposes different development modes for regional tourism. In practice, this study aims to identify the nonlinear functional mechanisms of the influential factors in regional tourism economic disparity, providing targeted guidance for regional tourism to enhance competitiveness and foster coordinated development.

## Literature review and asymmetric causality modeling

### Influential factors of regional tourism economic disparity

The influential factors of regional tourism economic disparity, as reported in the existing literature, include, but are not limited to, the local economy, tourism attraction, transportation

convenience, policy support, and geographic location. According to the hypothesis that the economy drives tourism growth, the local economy is essential for tourism development [19]. In the process of local economic expansion and growth, there would be more business and leisure travelers, better tourism-related infrastructure, and higher service quality; therefore, the overall competitiveness and development level of a tourist destination could be enhanced [20]. This hypothesis has been verified by many researchers, who arrived at three different conclusions regarding the relationship between the local economy and tourism growth: one-way causality [16, 21], two-way causality [22, 23], and no significant causality [24, 25]. Although the hypothesis could not always obtain evidence from practice [19, 26], it manifested a close relationship between the local economy and tourism development, proving that the local economy was an important factor in tourism economic disparities.

As a core element of tourism products, attractions play a significant role in local tourism development and largely determine the potential of local tourism [15, 19]. Local tourist attractions are crucial for attracting tourists because they provide essential recreational opportunities to satisfy travel motivations [7]. While it is widely believed that tourism attraction has a positive impact on local tourism development [3, 19], scholars like Deng, Ma, and Cao [27], as well as Seckelmann [28], have observed a mismatch, indicating that substantial tourism attraction does not necessarily lead to a high level of tourism economy, the issue of a "resource curse" also exists in tourism. Opinions vary regarding the impact of tourism attractions, but it is undeniable that they underpin tourism development [19]. Disparity in tourist attractions is a major factor that contributes to the unbalanced development of regional tourism.

Transportation convenience is a prerequisite for regional tourism development because it determines the accessibility of a tourist destination [29]. A sophisticated transportation infrastructure can increase the attractiveness of a tourist destination and improve tourists' experiences and satisfaction so that tourists are more likely to choose it [30, 31]. A convenient transportation network facilitates tourists' quick flow between their origins and destinations and increases the number of tourists and tourism revenue of the destinations, ultimately boosting the economic development of regional tourism [32, 33]. Substantial empirical research has verified the importance of transportation infrastructure in tourism development [29, 31, 32], so that transportation convenience is an important factor that influences regional tourism economic disparity.

Governmental policy support catalyzes the regional tourism economy. First, government attention to tourism is a driving force for regional tourism development. Publicity of a tourist destination is conducive to the establishment and expansion of the tourist market [34, 35]. Secondly, public service facility, a necessity for regional tourism development, was usually equipped by the government, thus favorable government policy could stimulate investment in regional tourism [36, 37]. Third, the government could ensure the orderly operation of regional tourism through industrial management policies so that policy support guarantees the sound and sustainable development of regional tourism during a crisis [35, 38]. Regional tourism development cannot occur without government policy support; therefore, differences in policy support lead to economic disparities in regional tourism.

Geographic location whose impact on tourism was often associated with spatial effect could not be neglected in regional tourism development [13, 19]. Spatial effects refer mainly to spatial spillovers and heterogeneity [19]. The former reflects the indirect influence exerted by a region's tourism development on tourist flow in its neighboring region. Such spillover is the result of spatial externalities among regions [39]. Spatial heterogeneity indicates that economic disparities in regional tourism are caused by inherent differences between regions [13]. Yang and Fik [19] demonstrate that considering spatial spillover and heterogeneity can significantly enhance the effectiveness of a regional tourism growth model. Geographically, the spatial effect

of China's tourism was quite evident [12, 13]. Academically, China is usually divided into eastern, central, and western regions. Their spatial heterogeneity was large, either in the natural environment or in socioeconomic development, while spatial homogenization within each region was relatively high, so spatial spillover was prominent [7, 39]. Consequently, both spatial heterogeneity and spatial homogenization are thought to be important for regional tourism economic disparities [7].

In addition to identifying influential factors, some studies have commented on their complex interactions [6, 13, 19]. Olya and Mehran [1] indicated that when predicting the influential variables of complex issues, a series of combined factors should be considered to gain deeper insight into the formation mechanism of the results. This conclusion also applies to the regional tourism economy, whose influential factors are heterogeneous and highly interactive. Empirical research on regional tourism development has revealed the heterogeneous functioning of the influential factors. Yang and Fik [19] adopted a geographically weighted Spatial Dubin Model to distinguish different development models of regional tourism and found that the local economy, tourism attractions, and other influential factors functioned heterogeneously. Jin et al. [13] analyzed regional tourism from the perspective of temporal and spatial heterogeneity and found that the local economy boosted regional tourism development more in developed regions than in less developed ones. This heterogeneity can be explained by the coexistence of influential factors [1]. Nevertheless, the current discussion on the causes of regional tourism economic disparity is mainly based on a regression model with symmetric causality; thus, the complex interaction of influential factors has not been revealed [13, 18]. Meanwhile, with traditional symmetric analysis, the occurrence of multicollinearity issues, non-normal datasets, and neglecting opposite cases and other factors in the control model can easily generate misleading results [40]. In summary, the regression model and symmetric analysis cannot fully clarify the complex causality of regional economic tourism disparity. Thus, a new perspective, such as the asymmetric approach, is needed to complement traditional symmetric analysis [1], as it could broaden the understanding of the complex causality underlying regional tourism economic disparities.

## Asymmetric causality modeling of regional tourism economic disparity

In accordance with the above review on antecedent variables of regional tourism economic disparity, this study selects the local economy, tourism attraction, transport convenience, policy support, and geographic location as the antecedent conditions for the asymmetric causality analysis of regional tourism economic disparity. To interpret the regional tourism economic disparity, this study utilizes a Venn Diagram to illustrate the proposed asymmetric causality model (Fig 1), depicting combinations of the five antecedent conditions based on the guidelines proposed by Dusa [41]. This model is asymmetric because it divides "disparity" into high and low levels of tourism economy, conducting an asymmetric analysis to identify the reasons for different levels. Antecedent conditions exert asymmetric effects, meaning that they may be positive, negative, or ineffective in different combinations. In addition, complex interactions exist between them, leading to the formation of specific situations that function only under certain conditions. Compared with the hypothesis of a unified causality effect in symmetric analysis, this model effectively illustrates asymmetric and complex causality.

## Research design and research method
### Variables and source of data

By referring to related research and considering the accessibility, representativeness, and comparability of the data, this study finalized specific measurement indicators for each antecedent

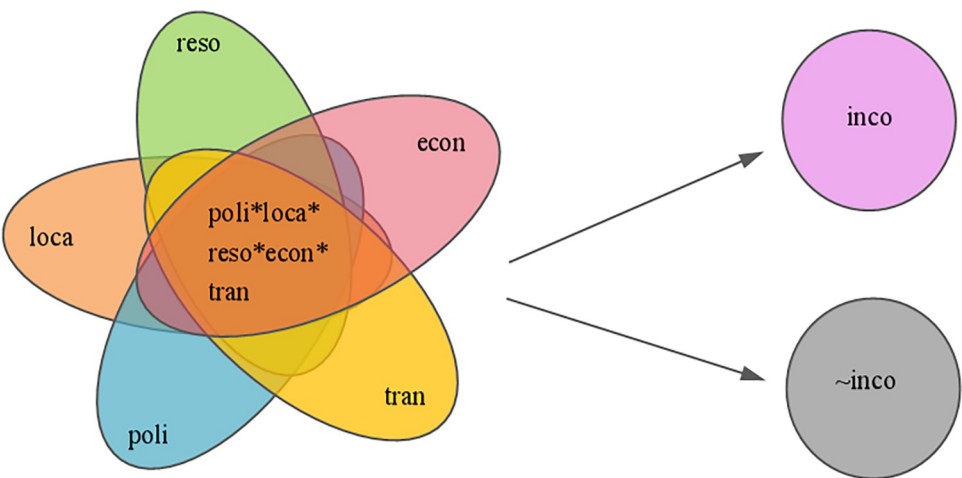

**Fig 1. Asymmetric Causality Model (econ, reso, tran, poli, loca and inco represent local economy, tourism attraction, transportation convenience, policy support, geographic location, and tourism income, respectively).**

variable. The local economy, which reflects the macroeconomic environment of a region, can be measured by the per-capita GDP of each province [42]. Tourism attraction largely determines the attractiveness of a tourist destination, so the magnitude of existing regional tourism attractions can be measured by the number of AAA and above-rated scenic areas (including all famous attractions such as World Heritage Sites and national parks) [19]. Transportation convenience was the precondition of tourists' entry so that it could be reflected by passenger-kilometers [43]. Policy support can be represented by the proportion of tourism income to GDP in each province because it generally signifies tourism's position in the overall industry structure through which tourism-related governmental policy support or even capital investment can be speculated [44]. Geographic location was used as a categorical variable. It is customary to analyze China in the eastern, central, and western regions [39]. As a result, the level of the regional tourism economy was measured as the sum of each province's domestic and inbound tourism income. Original data were obtained from the official websites of the National Bureau of Statistics (http://www.stats.gov.cn/), Ministry of Culture and Tourism (https://www.mct.gov.cn/), Provincial Bureau of Statistics, Provincial Department of Culture and Tourism, and the yearbooks of China's tourism statistics. Due to the impact of the COVID-19 pandemic, data from the most recent three years may not accurately reflect normal tourism development. Therefore, data from 2005 to 2019 were used to establish a database for this study.

## fsQCA and data analysis

To determine the causal mechanism for China's regional tourism economic disparity asymmetrically, this study applies fsQCA, which is a combination of fuzzy logic and qualitative comparative analysis (QCA) and a technique that integrates quantitative empirical tests and qualitative inductive reasoning [45, 46]. The foundations of QCA are set theory and Boolean algebra. With QCA, the causality of complex issues can be constructed with a small sample so that they can be easily comprehended using set theory [47, 48]. QCA has three major advantages: low requirement for sample size so that complex causality could be revealed with a small sample; equivalence, which means that multiple approaches to the results could exist concurrently and different combinations or sets of conditions could lead to the same result; and asymmetric causality, which means that the occurrence and non-occurrence of a result should

be explained, respectively [47, 49]. Using fuzzy set theory and Boolean algebra, fsQCA can determine how to combine these antecedent conditions and what results will be obtained with or without them.

Calibration and calculation of the fuzzy truth table are the main steps in fsQCA. Calibration refers to the membership scores of antecedent and result variables when their values are converted into a fuzzy set. For this purpose, all variables were designated with three qualitative anchor points to describe the extent to which they belonged to a set; specifically, 1 indicated a full membership set, 0.5 indicated an intersection, and 0 indicated non-membership [47, 50]. Mathematically, membership scores of 1 and 0 could not be obtained in this research; therefore, on the basis of the threshold values for full membership and non-membership proposed by Ragin [47], which stood at 0.95 and 0.05 respectively, the maximum value of a variable was determined to be the threshold value of full membership, the minimum value was the threshold of non-membership, and the median was the threshold for the intersection. The truth table is a list of antecedent condition combinations that formulate all possible configurations of the predicted results. Configuration conditions should be refined using frequency and consistency standards [1]. To predict the level of tourism economy, the value 1 was regarded as a division point for frequency, and 0.8 is the unanimous threshold value for consistency [48]. The degree of consistency and coverage of the fsQCA provides precise information on how causality configurations interpret specific results. Their measurements were similar to those of the correlation and determination coefficients in symmetric methods. Consistency can be calculated using $(X_i \leq Y_i) = \sum\{\min(X_i, Y_i)\}/\sum(X_i)$, whereas the formula for the coverage degree is $(X_i \leq Y_i) = \sum\{\min(X_i, Y_i)\}/\sum(Y_i)$. $X_i$ is the membership score of case i in set X and $Y_i$ is the membership score of case i under the result variable [47].

In general, the data analysis of this research was conducted in the following four steps: Step 1—the measurement of regional tourism economic disparity through standard deviation (SD), coefficient of variation (CV), and gradient of tourism economy level (T). SD and CV indicate the absolute and relative deviations of the regional tourism economy, respectively, and the gradient value of the level of the tourism economy specifies different levels of the regional tourism economy. The formula is as follows:

$$\mathrm{CV} = \sqrt{\sum_{i=1}^{31}(y_i - \bar{y})^2/31}/\bar{y}$$

$$\mathrm{SD} = \sqrt{\sum_{i=1}^{31}(y_i - \bar{y})^2/31}$$

$$\mathrm{T} = y_i/\bar{y}$$

$y_i$ is the total tourism income of province i in a certain year, and $\bar{y}$ is the mean value of tourism income of the 31 provinces in that year. To directly reflect regional tourism economic disparity and its evolution tendency, this study calculates the standard deviation and variation coefficient of each province's total tourism income from 2005 to 2019, but calculates the gradient of the tourism economy level with the mean value of data from 2014 to 2019 to reduce the impact of normal economic fluctuations and focus more on the current situation. Given that regional tourism economic disparity is an unquestioned fact and has been measured in previous research, this study only provides the necessary background information and an explanation of its causal mechanism.

Step 2: The variable relationship plot is applied to reveal the opposite cases. This step aims to justify fsQCA-based asymmetric causality modeling by explaining the shortcomings of traditional symmetric analysis. Step 3: Asymmetric modeling with fsQCA targets exploring the causality configuration of the results (different levels of the tourism economy). Step 4: The prediction validity analysis verifies the prediction validity of the model using other samples. Statistical data from 2014 to 2019 were adopted to calculate each province's level of tourism economy and the mean value of antecedent variables, which were further analyzed using fsQCA 3.0.

## Result and analysis

### Measurement of regional tourism economic disparity

According to Fig 2, the regional tourism economic disparity is long in China, but the absolute and relative deviations are different. From 2005 to 2019, the absolute deviation increased from 541.79 to 4161.41, increasing by 15.77% annually, whereas the relative deviation dropped from 0.94 to 0.59, decreasing by 3.26% annually. In other words, the absolute deviation of the regional tourism economic disparity continues to expand, whereas its relative deviation slowly decreases. A possible reason for this is that in the rapid development course of China's tourism, the overall tourism economy continues to grow. However, regions have distinct economic foundations, tourist attractions, and geographical locations. Consequently, the development foundation and growth rate of the tourism economy vary across regions, leading to inconsistent patterns, and economic disparity in regional tourism is increasing. To find specific approaches to the economic growth of regional tourism, it is important to analyze the reasons for disparity under certain antecedent conditions.

By calculating the gradient value of each province's level of tourism economy (T), this research classifies the 31 provinces into four types: high level ($1.5 \leq T$), relatively high level ($1 \leq T < 1.5$), relatively low level ($0.5 \leq T < 1$), and low level ($0 < T < 0.5$), as shown in Table 1. Eighteen provinces were at or above a relatively high level, among which Guangdong, Jiangsu,

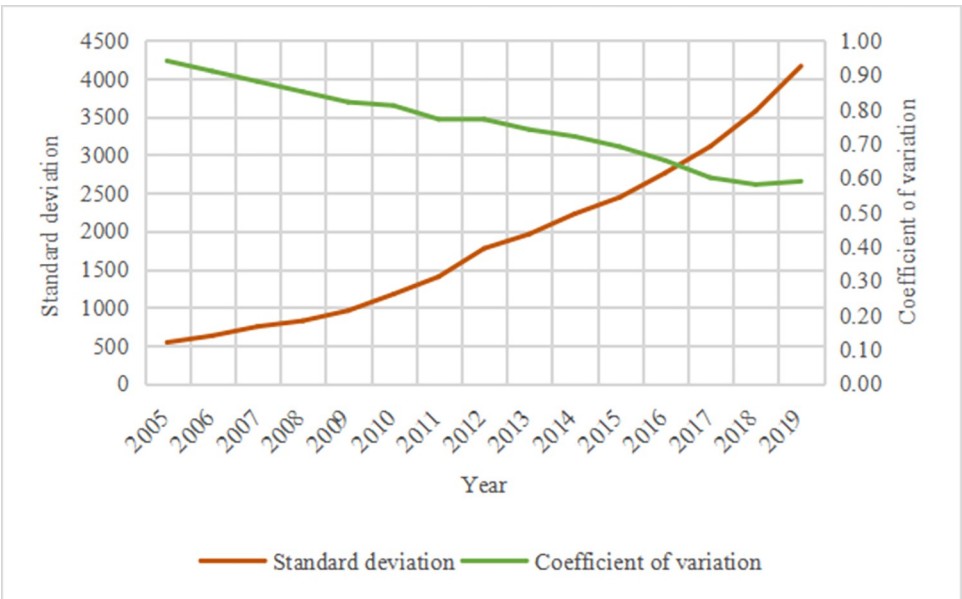

**Fig 2. China's provincial tourism economic disparity from 2005 to 2019.**

**Table 1. Tourism economy level of 31 provinces in Mainland China.**

| Level of tourism economy | Provinces |
|---|---|
| High level | Guangdong, Jiangsu, Shandong, Zhejiang, Sichuan |
| Relatively high level | Guizhou, Henan, Yunnan, Hunan, Jiangxi, Anhui, Hebei, Guangxi, Hubei, Beijing, Fujian, Shanxi, Liaoning |
| Relatively low level | Shaanxi, Shanghai, Jilin, Chongqing, Tianjin, Inner Mongolia |
| Low level | Xinjiang, Heilongjiang, Gansu, Hainan, Tibet, Qinghai, Ningxia |

Shandong, Zhejiang, and Sichuan were at a high level. Thirteen provinces were at or below a relatively low level, including Heilongjiang, Xinjiang, Gansu, Hainan, Tibet, Qinghai, and Ningxia are low level. The combinations of antecedent conditions that lead to high/low levels of the tourism economy were analyzed for feasible solutions to each province's tourism economy.

## Analysis of opposite cases

Fig 3 indicates that the relationship between antecedent conditions and the level of the tourism economy is not simply positive or negative; in fact, cases that are opposite to the major influence relationship also exist. Taking the relationship between the local economy and the level of tourism economy as an example, it is positive in general, that is, a high local economy level brings about a high level of tourism economy, and a low local economy causes a low level of tourism economy, but there are opposite cases, which means provinces with low economic levels have a high level of tourism economy, and provinces with high economic levels have a low level of tourism economy. To clarify this phenomenon, 15, 14, 8, and 3 opposite cases are marked in Fig 3A–3D, respectively. To include these opposite cases in the prediction of high/low levels of the tourism economy, fsQCA was used to conduct data analysis.

## fsQCA results

As shown in Table 2, three models (M1-1, M1-2, and M1-3), constituted by different combinations of the five antecedent conditions, present a high level of tourism economy, which means that there are three sufficient and consistent approaches to a relatively high level of tourism economy. Ragin [47] defined that the critical values of coverage and consistency should be higher than 0.20 and 0.75, respectively, so M1 has ideal coverage (0.65) and consistency (0.95).

In M1-1 (loca*~poli*econ*reso*tran), provinces with favorable geographic locations, local economies, tourism attractions, and transportation convenience, but insufficient policy support enjoy a high level of tourism economy. Among the provinces whose membership score surpasses 0.5, Jiangsu, Zhejiang, Shandong, and Guangdong have a high level of tourism economy, and their complex causality condition is the same as that of the antecedent combination in M1-1. According to Fig 4A, the relationship between X (M1-1, a model of antecedent combination) and Y (level of tourism economy) is asymmetric or sufficient, but not necessary. That is, the combination of antecedent conditions (X) is a sufficient (but not necessary) and consistent complex causality condition for a high-level tourism economy (Y).

Similarly, M1-2 displays an asymmetric relationship between X and Y, as illustrated in Fig 4B. Therefore, M1-2 (~loca*poli*~econ*reso*tran) is also a sufficient and consistent solution for a high-level tourism economy. Specifically, in provinces where geographic location and the local economy are poor, policy support, tourism attraction, and transportation convenience are advantageous, and a relatively high level of tourism economy is possible. Typical examples include Guangxi, Sichuan, and Guizhou. Another approach to a high-level tourism

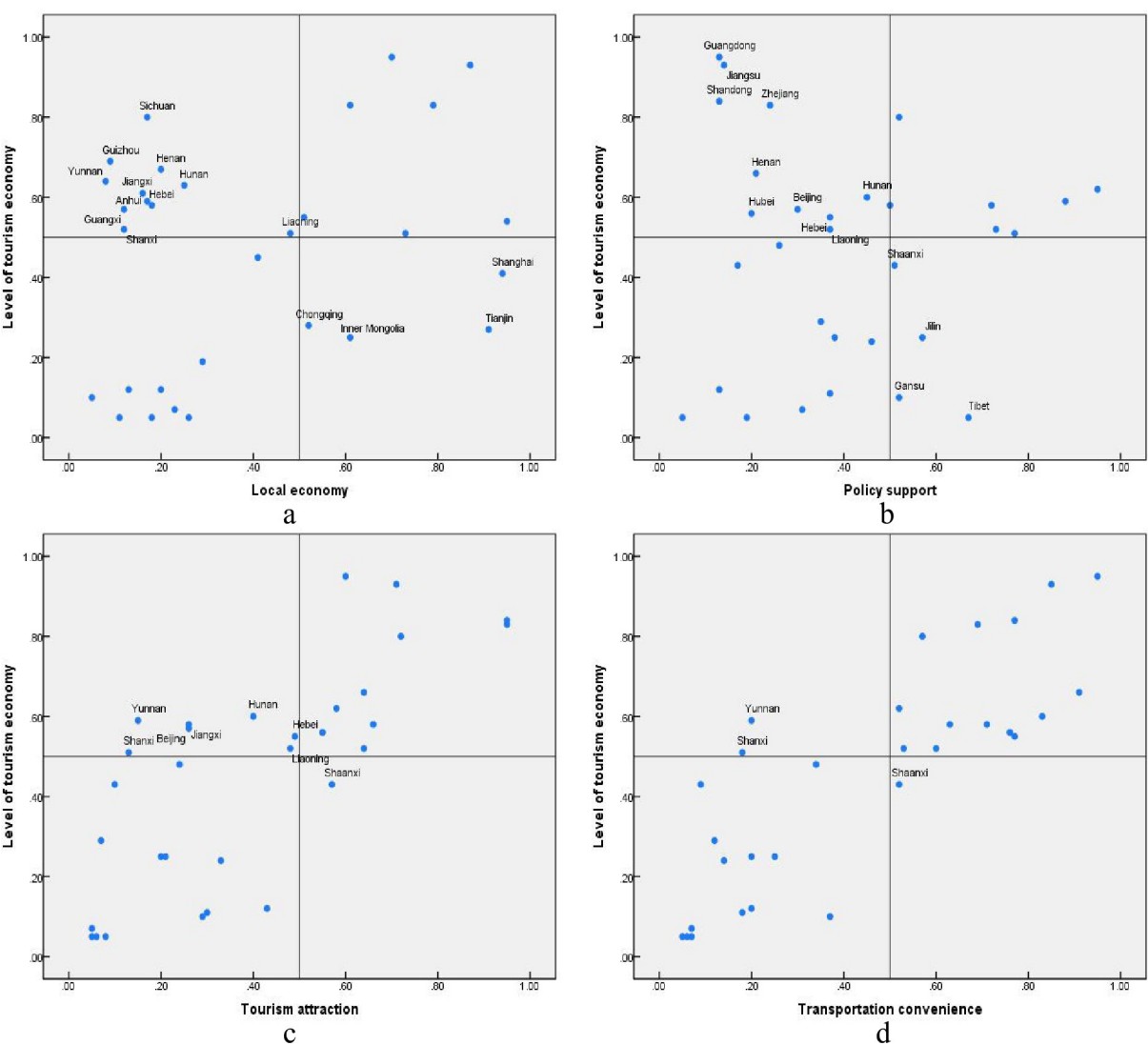

**Fig 3. Relationship between different antecedent conditions and level of tourism economy.**

economy is M1-3 (loca*~poli*~econ*~reso*tran), which indicates that in provinces where policy support, local economy, and tourism attraction are poor but geographic location and transportation convenience are good, relatively high levels of tourism economy are possible.

**Table 2. Complex configuration of antecedent conditions in predicting high level of tourism economy.**

| Causality model for high level of tourism economy | Raw coverage | Unique coverage | Consistency |
|---|---|---|---|
| M1: inco = f (loca, poli, econ, reso, tran) | | | |
| M1-1: loca*~poli*econ*reso*tran | 0.406 | 0.150 | 0.998 |
| M1-2: ~loca*poli*~econ*reso*tran | 0.370 | 0.168 | 0.951 |
| M1-3: loca*~poli*~econ*reso*tran | 0.335 | 0.052 | 0.962 |
| Solution coverage: 0.653 | | | |
| Solution consistency: 0.954 | | | |

Note: In the logical calculus of Boolean algebra, "Negation" is represented by "~"; "And" is represented by "*".

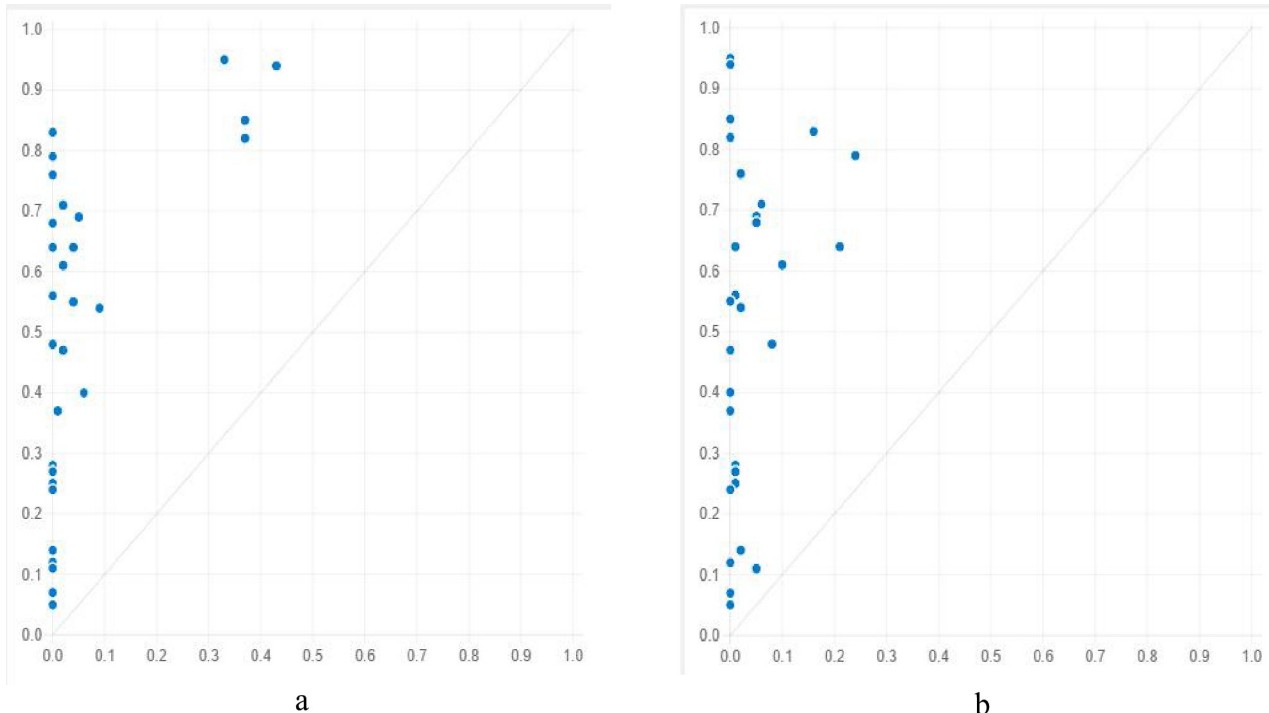

**Fig 4.** XY graph of Causality Models for High Level of Tourism Economy (a: XY graph of M1-1; b: XY graph of M1-2).

Unlike in traditional research, some antecedent variables of regional tourism economic disparity have positive and negative impacts on the three models; therefore, it is important to explain their complexity and heterogeneity. Jang and Ham [51] noted that the function of an indicator is determined by other causal antecedents (presence, absence, or magnitude) in the model. For example, the local economy has a positive impact in M1-1, but a negative impact in M1-2 and M1-3; tourism attraction has a positive impact in M1-1 and M1-2, but a negative impact in M1-3. Only transportation convenience functions in the same manner in every combination of antecedent conditions and is not influenced by other variables. Therefore, this is a necessary condition for a high tourism economy. Ordanini, Parasuraman, and Rubera [49] concluded that the causality configuration for result prediction was more important than predicting antecedent variables.

According to Table 3, three models (M2-1, M2-2, and M2-3) predict a low tourism economy. Coverage (0.86) and consistency (0.86) were satisfactory. Notably, these three models are not exactly the opposite of the models for a high-level tourism economy. For example, M2-1 (~poli*~reso*~tran) demonstrates that disadvantageous policy support, tourism attractions,

**Table 3. Complex configuration of antecedent conditions in predicting low level of tourism economy.**

| Causality model for low level of tourism economy | Raw coverage | Unique coverage | Consistency |
|---|---|---|---|
| M2: ~inco = f (loca, poli, econ, reso, tran) | | | |
| M2-1: ~poli*~reso*~tran | 0.713 | 0.141 | 0.936 |
| M2-2: ~loca*~econ*~reso*~tran | 0.574 | 0.091 | 0.903 |
| M2-3: loca*~poli*~econ*~reso | 0.353 | 0.053 | 0.888 |
| Solution coverage: 0.858 | | | |
| Solution consistency: 0.856 | | | |

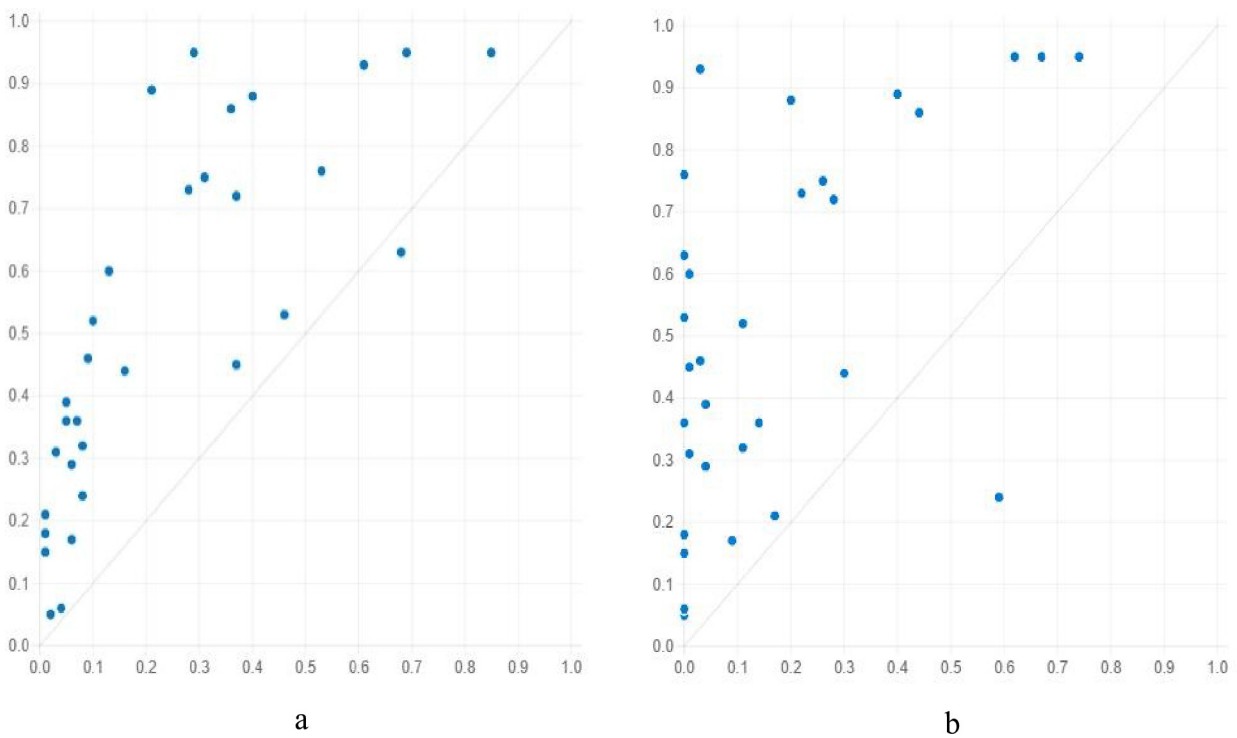

**Fig 5.** XY graph of Causality Models for Low Level of Tourism Economy (a: XY graph of M2-1; b: XY graph of M2-2).

and transportation convenience result in a low-level tourism economy. Typical examples include Inner Mongolia, Heilongjiang, Chongqing, and Hainan. M2-2 (~loca*~econ*~reso*~-tran) describes another combination of antecedent conditions for low levels of the tourism economy, that is, bad geographic location, an underdeveloped local economy, poor tourism attraction, and low transportation convenience, similar to the situation in Gansu, Ningxia, Qinghai, and Tibet.

According to Fig 5, the relationship between the causality model and the low level of the tourism economy is also asymmetric or sufficient, but not necessary. Therefore, the combinations of the antecedent conditions in these three models are sufficient for a low-level tourism economy. M2-3 (loca*~poli*~econ*~reso) is another causality model for low levels of the tourism economy, which means that the level of the tourism economy in provinces with good geographic location but insufficient policy support, local economy, and tourism attraction is low.

Overall, the antecedent variables are heterogeneous and complex in M2 because some variables have a negative, positive, or no impact on different combinations. For example, geographic location is not included in M2-1, is negative in M2-2, and is positive in M2-3. Policy support is not included in M2-2 and is negative in M2-1 and M2-3. The only exception is tourism attraction, which functions similarly in each combination, indicating that a poor tourism attraction is a necessary condition for a low-level tourism economy.

## Prediction validity

One-time favorable model fitting does not guarantee good predictive ability; therefore, prediction validity is employed to assess the predictive capability of the hypothetical asymmetric causality model across different datasets. Following Olya and Mehran's [1] approach, the original data were divided into two subsamples. Subsample 1 underwent fsQCA for asymmetric

**Table 4. Result of prediction validity analysis.**

| Causality model for high level of tourism economy with subsample 1 | Raw coverage | Unique coverage | Consistency |
|---|---|---|---|
| Model: inco = f (loca, poli, econ, reso, tran) | | | |
| M1: loca*~poli*econ*reso*tran | 0.390 | 0.132 | 0.997 |
| M2: ~loca*poli*~econ*reso*tran | 0.293 | 0.111 | 0.986 |
| M3: loca*~poli*~econ*~reso*tran | 0.342 | 0.072 | 0.969 |
| Solution coverage: 0.586 | | | |
| Solution consistency: 0.977 | | | |

causality modeling. Subsequently, the causality models derived from subsample 1 were tested using subsample 2.

Table 4 shows that antecedent condition combinations acquired from fsQCA modeling for high level of tourism economy with subsample 1 is the same with those of the whole sample, and its coverage (0.59) and consistency (0.98) are good. Specifically, Fig 6A reveals an asymmetric relationship of sufficient but not necessary between X (M1, a model of antecedent condition combination) and Y (level of tourism economy). Then, subsample 2 is utilized to test the validity of M1, a model gained with subsample 1. As illustrated by Fig 6B, there are similar asymmetric relationship, coverage (0.72) and consistency (0.94). In conclusion, the asymmetric causality model is valid in prediction.

## Discussion and conclusion

### Discussion

In contrast to previous symmetric perspectives and models, this study innovatively reveals the complex causal mechanism of regional tourism economic disparity through asymmetric

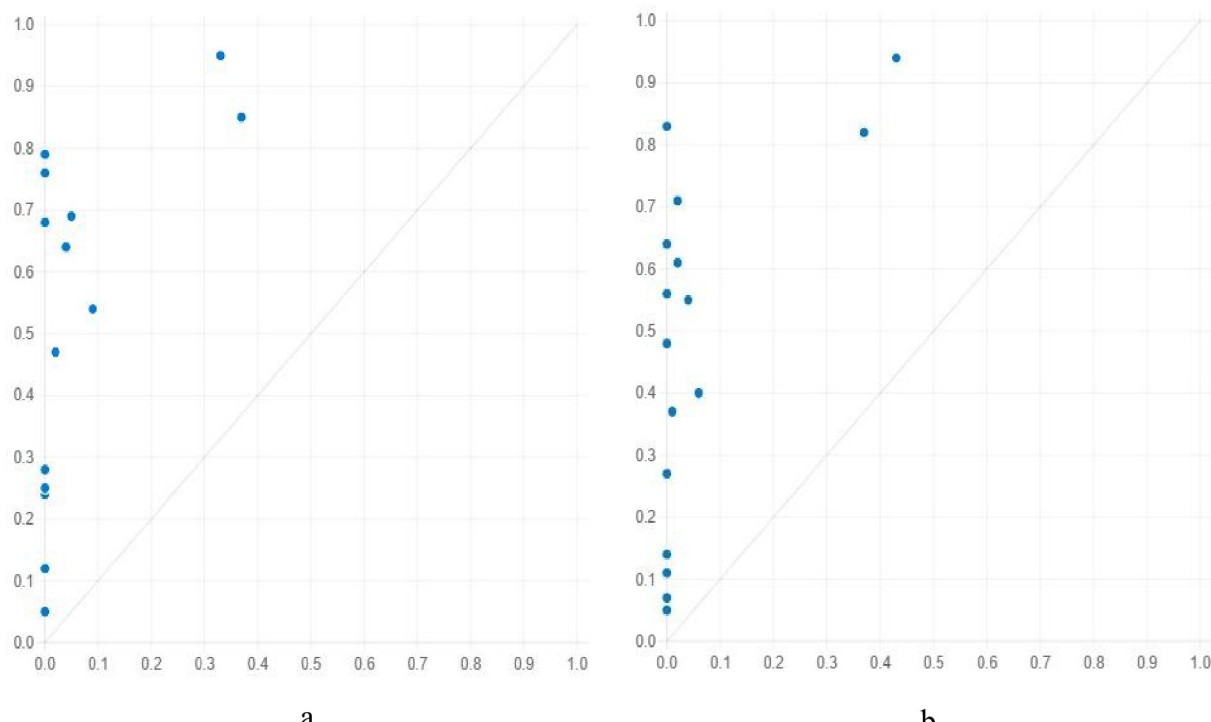

**Fig 6.** XY Graphs of M1 with Different Subsamples (a: XY graph of M1 with subsample 1; b: XY graph of M1 with subsample 2).

modeling. Using fsQCA, an asymmetric model for the complex causation of the disparity is constructed to obtain three antecedent condition combinations for the prediction of a high-level tourism economy and another three for a low-level tourism economy. The former three combinations are not exactly opposite to the latter three combinations; therefore, the two sides cannot be applied to explain regional tourism disparity. However, in traditional symmetric research, since antecedent variables exert either positive or negative effects on result variables, the opposite of the reasons for a high level of tourism economy is often used to explain a low level of tourism economy [6, 18].

Among the antecedent variables of regional tourism economic disparity, most present asymmetric nonlinear effects, as they vary with the existence and influence direction of other antecedent variables. For example, in the prediction model for a high-level tourism economy, the effect of the local economy will be influenced by other variables, so the local economy does not impact the tourism economy level in a linear way, but impacts it positively or negatively depending on the changes in other variables in the model. This result helps us understand the nonlinear relationship between the local economy and tourism development [11, 19]. Tourism attractions function differently in different combinations of antecedent conditions; therefore, they are also impacted by other antecedent variables. This could be one reason for the resource curse [27]; the issue of whether the resource curse exists should be considered with other factors such as economy, transportation, policy, and geographic location because they constitute situational conditions together.

This study also reveals that good transportation convenience is a necessary condition for a high-level tourism economy, whereas poor tourism attraction is a necessary condition for a low-level tourism economy. This aligns with conclusions drawn from symmetric research, indicating that better transportation is conducive to the improvement of the tourism economy [31, 33], and regions with low economic value in tourism usually lack tourism attractions [13, 14].

## Conclusion

This study constructs an asymmetric causality model for regional tourism economic disparity and verifies its validity using a prediction validity analysis. This framework explains regional economic disparities in tourism. Using fsQCA asymmetric modeling, six combinations of antecedent conditions that predict the level of the tourism economy were acquired, insights on how those variables interactively impact economic disparity were gained, and development modes for the regional tourism economy were explored.

This study makes a significant contribution by shedding light on the complex causality of regional tourism economic disparities through a new perspective on the relationship between these disparities and their influencing factors. The different effects of these influential factors are no longer controversial given that certain factors may function differently under different conditions. In addition, when discussing a factor's functions, it is necessary to combine other common factors to distinguish between functioning situations. These findings supplement those of symmetric research; thus, they can expand the perception of the complex interaction among the influential factors of regional tourism economic disparity and their nonlinear functional mechanism.

Tourism policymakers can obtain practical guidance from this study. First, the six combinations of antecedent conditions acquired from asymmetric modeling provide multiple approaches for the economic development of regional tourism so that different regions can select suitable solutions accordingly. Second, regarding regional tourism development, the configuration of influential factors should be considered to avoid overinvesting in a single factor. For example, policy support works well in places where the local economy and geographic

location are poor but tourism attractions and transportation convenience are relatively good. However, in the other situations, this effect was not significant. Third, against the background of deepening tourism development, continuous investment in transportation infrastructure and upgraded transportation networks can fully tap into the potential of transportation to promote the tourism economy. To narrow down the regional tourism economic disparity, regions with low levels of tourism economy need to reverse their disadvantages in tourism attractions by intensifying the development of tourism resources, particularly high-quality resources, and promoting the innovative transformation and development of cultural and tourism resources. In the meantime, tourist attractions and the corresponding transportation facilities should be perfected together to advance the regional tourism economy and bring more economic benefits.

This study had some limitations that necessitate further study. For instance, the number of cases and conditions considered in fsQCA should be maintained at an appropriate proportion. The purpose of QCA is not to list all variables in a case, but to discover or approach the reasons for a certain phenomenon by analyzing the consistency of important variables. In this study, 31 provinces in mainland China were selected as cases, with five major antecedent conditions serving as explanatory variables. Subsequent research could delve into the asymmetric causality between regional tourism economic disparity and other influential factors to gain a deeper understanding of this complex phenomenon. The formation mechanism of regional tourism economic disparity may change over different periods, making an asymmetric analysis of its evolution at different stages a potential research direction. Additionally, this research demonstrates the applicability of fsQCA in analyzing the asymmetric causality of regional tourism economic disparity. In reality, it can be employed to analyze tourism economic disparity at other spatial scales and explore other topics related to the tourism economy.

## Acknowledgments

We extend our sincere gratitude to the reviewing experts and editors for their valuable suggestions and meticulous review, which have significantly enhanced the quality of this manuscript. We would like to thank Editage (www.editage.com) for English language editing.

## Author Contributions

**Conceptualization:** Huan Ling, Xiaoyue Qiu.

**Formal analysis:** Huan Ling.

**Methodology:** Huan Ling, Xiaoyue Qiu.

**Writing – original draft:** Huan Ling.

**Writing – review & editing:** Huan Ling, Xiaoyue Qiu.

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
