## [Decision Letter · Decision Letter 0]

16 Jan 2024

PONE-D-24-00500Asymmetric modeling of regional tourism economic disparity in ChinaPLOS ONE

Dear Dr. Ling,

Thank you for submitting your manuscript to PLOS ONE. After careful consideration, we feel that it has merit but does not fully meet PLOS ONE’s publication criteria as it currently stands. Therefore, we invite you to submit a revised version of the manuscript that addresses the points raised during the review process.

We look forward to receiving your revised manuscript.

Kind regards,

Saliha Karadayi-Usta, PhD

Academic Editor

PLOS ONE

Journal Requirements:

4. We note that Figure 3 in your submission contain map/satellite images which may be copyrighted. All PLOS content is published under the Creative Commons Attribution License (CC BY 4.0), which means that the manuscript, images, and Supporting Information files will be freely available online, and any third party is permitted to access, download, copy, distribute, and use these materials in any way, even commercially, with proper attribution. For these reasons, we cannot publish previously copyrighted maps or satellite images created using proprietary data, such as Google software (Google Maps, Street View, and Earth). For more information, see our copyright guidelines: http://journals.plos.org/plosone/s/licenses-and-copyright.

a. You may seek permission from the original copyright holder of Figure 3 to publish the content specifically under the CC BY 4.0 license.  

Additional Editor Comments:

Please take the below reviewer suggestions into consideration and make necessary revisions.

Reviewers' comments:

Reviewer's Responses to Questions

**Comments to the Author**

1. Is the manuscript technically sound, and do the data support the conclusions?

Reviewer #1: Yes

Reviewer #2: Yes

Reviewer #3: Yes

2. Has the statistical analysis been performed appropriately and rigorously? 

Reviewer #1: Yes

Reviewer #2: Yes

Reviewer #3: Yes

3. Have the authors made all data underlying the findings in their manuscript fully available?

Reviewer #1: Yes

Reviewer #2: Yes

Reviewer #3: Yes

4. Is the manuscript presented in an intelligible fashion and written in standard English?

Reviewer #1: No

Reviewer #2: Yes

Reviewer #3: Yes

5. Review Comments to the Author

Reviewer #1: I have reviewed the article titled "Asymmetric Modeling of Regional Tourism Economic Disparity in China," and I am pleased to provide you with a positive review.

The article delves into an important and often overlooked aspect of China's tourism economy – the distinctive regional disparity. The use of statistical data from 31 provinces in mainland China showcases the comprehensive nature of the research. The application of fuzzy set qualitative comparative analysis (fsQCA) adds a unique dimension, allowing for a nuanced understanding of the complex interactions that contribute to regional tourism economic disparities.

One of the strengths of the article lies in its identification and exploration of asymmetric causes behind regional tourism economic disparity. The differentiation between high and low levels of tourism economy formation approaches adds depth to the analysis. The recognition of the non-linear effects of influential factors, influenced by antecedent conditions, is a valuable contribution to the field.

The identification of transportation convenience as a necessary condition for a high level of tourism economy and poor tourism attraction as a necessary condition for a low level of tourism economy is a significant finding. This insight not only broadens our understanding of the determinants of regional tourism economic performance but also provides practical guidance for policymakers.

Overall, the article not only offers a fresh perspective on regional tourism economic disparity but also contributes to the development of a region-based tourism model. The research is well-executed, providing valuable insights for both academics and policymakers in the tourism sector.

Suggestions:

1. Add more citations.

E.g.

https://www.mdpi.com/2071-1050/15/13/10398

and

https://www.frontiersin.org/articles/10.3389/fevo.2023.1148868/full

Etc.

2. Native language correcrtion are needed.

Reviewer #2: The authors need to mention (at least, sometimes) the scholars instead of mere using digits. Otherwise, we need to turn pages up and down more than 40 times to see who the scholar was.

Bibliography rises a question: what sort of references have been used. It is not MLA, Chicago, APA, Turabian. E.g., Pappas IO, Woodside AG. Should have been Pappas, I.O., Woodside, A.G.

Page 5 line 106: …. literature include but (Passive voice is necessary) not limited to local economy

Line 248: comprehended with set theory[45, 46] (a blank space is necessary)

Line 257: Calibration is refers to the membership (please check the grammar)

Line 313-315 But different regions have (we suggest adding here “different”) economic foundation, tourism attraction, and geographic location, consequently, ….

Fig 1, 2 and others should include a period, e.g., Fig. 1, Fig. 2 (check through the entire text)

Line 481 I would rather be careful declaring “The biggest contribution of this research lies in revealing the complex causality….” because the authors’ conclusion does not obligatorily mean the reviewers or large audience’s opinion or conclusion.

Reviewer #3: The originality of this paper is the utilizing statistical data from the tourism economy in 31 provinces in mainland China, this research explores regional tourism economic disparity through asymmetric modeling of fuzzy set qualitative comparative analysis (fsQCA).

The theorectical contributions is the exploration of the causes of regional tourism economic disparity are asymmetric; high and low levels of tourism economy have different formation approaches; the functioning of the most influential factors is impacted by other antecedent conditions, presenting an asymmetric non-linear effect; and favorable transportation convenience is a necessary condition for a high level of tourism economy, while poor tourism attraction is a necessary condition for a low level of tourism economy.

Managerial implications of this research introduces a new perspective on regional tourism economic disparity, enabling in-depth comprehension of the complex interaction and non-linear functional mechanisms of influential factors. Besides, the authors explored a region-based tourism development model. Therefore, it can offer significant practical guidance for tourism policymakers.

6. PLOS authors have the option to publish the peer review history of their article (what does this mean?). If published, this will include your full peer review and any attached files.

Reviewer #1: No

Reviewer #2: **Yes: **Dr. Kamo Chilingaryan

Reviewer #3: **Yes: **Cuong Nguyen

---

## [Author Response · Author response to Decision Letter 0]

7 Feb 2024

Distinguished Editors and Reviewers,

Thank you for offering valuable revision suggestions for our manuscript titled “Asymmetric modeling of regional tourism economic disparity in China” (ID: PONE-D-24-00500). We are grateful for the efficient evaluation of our paper and the valuable opportunity provided for revision. Your insightful suggestions have significantly contributed to enhancing the quality of our work. We deeply appreciate and acknowledge your expertise, dedicating ourselves wholeheartedly to addressing each comment. We sincerely hope that these revisions meet with your approval. Modified contents are highlighted in yellow in the revised manuscript with track changes, and detailed responses to each suggested modification are provided below.

Response to journal requirements

1. Please ensure that your manuscript meets PLOS ONE’s style requirements, including those for file naming. The PLOS ONE style templates can be found at 

Response: Thank you for the guidance regarding PLOS ONE’s style requirements. We have meticulously reviewed and revised our manuscript, ensuring it aligns with the formatting guidelines outlined in the provided PLOS ONE style templates. The necessary adjustments have been made to meet the journal’s standards for file naming and overall formatting.

The name of the colleague or the details of the professional service that edited your manuscript

A copy of your manuscript showing your changes by either highlighting them or using track changes (uploaded as a *supporting information* file)

A clean copy of the edited manuscript (uploaded as the new *manuscript* file)

Response: Thank you for your valuable suggestion regarding language usage, spelling, and grammar. Taking this seriously, we have enlisted the services of Editage for a comprehensive edit of our manuscript. The necessary documents, including a copy of the manuscript showing changes and a clean edited version, have been uploaded as per your instructions.

Response: Thank you for highlighting this issue. We have thoroughly reviewed and updated the grant numbers in the ‘Funding Information’ section as per your guidance.

4. We note that Figure 3 in your submission contain map/satellite images which may be copyrighted. All PLOS content is published under the Creative Commons Attribution License (CC BY 4.0), which means that the manuscript, images, and Supporting Information files will be freely available online, and any third party is permitted to access, download, copy, distribute, and use these materials in any way, even commercially, with proper attribution. For these reasons, we cannot publish previously copyrighted maps or satellite images created using proprietary data, such as Google software (Google Maps, Street View, and Earth). For more information, see our copyright guidelines: http://journals.plos.org/plosone/s/licenses-and-copyright.

a. You may seek permission from the original copyright holder of Figure 3 to publish the content specifically under the CC BY 4.0 license.  

Response: Thank you for highlighting the concern about Figure 3. We would like to clarify that the base map in Figure 3 was sourced from the open and freely accessible standard map service provided by the Ministry of Natural Resources of the People’s Republic of China (http://bzdt.ch.mnr.gov.cn/), with no copyright disputes. However, to simplify the matter and ensure compliance, we have decided to remove Figure 3 from the submission. Instead, we have opted for a tabular representation, preserving the relevant information for clarity and comprehension. We hope this modification aligns with the journal’s requirements, and we greatly appreciate your understanding and approval.

Response: Thank you for your guidance on the reference list. We have thoroughly reviewed our reference list, ensuring its completeness and correctness. We used the EndNote tool to meticulously format the references in strict adherence to PLOS ONE guidelines. All necessary changes have been accurately reflected as per your instructions. We appreciate your attention to this matter.

Response to reviewers’ comments

Reviewer #1: I have reviewed the article titled “Asymmetric Modeling of Regional Tourism Economic Disparity in China,” and I am pleased to provide you with a positive review.

The article delves into an important and often overlooked aspect of China's tourism economy - the distinctive regional disparity. The use of statistical data from 31 provinces in mainland China showcases the comprehensive nature of the research. The application of fuzzy set qualitative comparative analysis (fsQCA) adds a unique dimension, allowing for a nuanced understanding of the complex interactions that contribute to regional tourism economic disparities.

One of the strengths of the article lies in its identification and exploration of asymmetric causes behind regional tourism economic disparity. The differentiation between high and low levels of tourism economy formation approaches adds depth to the analysis. The recognition of the non-linear effects of influential factors, influenced by antecedent conditions, is a valuable contribution to the field.

The identification of transportation convenience as a necessary condition for a high level of tourism economy and poor tourism attraction as a necessary condition for a low level of tourism economy is a significant finding. This insight not only broadens our understanding of the determinants of regional tourism economic performance but also provides practical guidance for policymakers.

Overall, the article not only offers a fresh perspective on regional tourism economic disparity but also contributes to the development of a region-based tourism model. The research is well-executed, providing valuable insights for both academics and policymakers in the tourism sector.

Response: Many thanks to the reviewing expert for the positive review of our article. We sincerely appreciate your thorough evaluation and commendation. Your recognition of the importance of our research and the comprehensive nature of the study is genuinely appreciated and serves as great encouragement for the authors.

Here are the specific responses to your suggestions:

1.Add more citations.

Response: Thank you for your meticulous recommendations regarding relevant literature. We have conscientiously studied and found inspiration in the suggested articles, incorporating them appropriately into our manuscript. For detailed citations, please refer to the references [8] and [15] in the main text:

[8] Zhang Y. The sustainability of regional innovation in China: insights from regional innovation values and their spatial distribution. Sustainability. 2023;15(13). doi: 10.3390/su151310398.

[15] Zhu K, Zhou Q, Cheng Y, Zhang Y, Li T, Yan X, et al. Regional sustainability: pressures and responses of tourism economy and ecological environment in the Yangtze River basin, China. Frontiers in Ecology and Evolution. 2023;11. doi: 10.3389/fevo.2023.1148868.

2.Native language correction is needed.

Response: Thank you for the suggestion, and we have taken your advice seriously. To ensure the highest quality, we engaged the professional editing services of Editage for a comprehensive language edit of our manuscript. Your thoughtful guidance has been instrumental in refining the clarity and precision of our work, and we are grateful for your expertise.

Reviewer #2: The authors need to mention (at least, sometimes) the scholars instead of mere using digits. Otherwise, we need to turn pages up and down more than 40 times to see who the scholar was.

Response: Thank you for the valuable suggestion from the reviewing expert. We sincerely apologize for any inconvenience caused by the previous lack of clarity. In response to your feedback, we have revised the manuscript to include scholars’ names alongside citation numbers. For example, we now mention “scholars like Deng, Ma, and Cao [27], as well as Seckelmann [28], have observed a mismatch, indicating that substantial tourism attraction does not necessarily lead to a high level of tourism economy” (Page 6, line 114); “Yang and Fik [19] demonstrate that considering spatial spillover and heterogeneity can significantly enhance the effectiveness of a regional tourism growth model” (Page 7, line 145); “Following Olya and Mehran’s [1] approach, the original data were divided into two subsamples” (Page 19, line 382), etc. Throughout the manuscript, we have made these modifications for clarity and ease of reference. We hope these changes enhance the readability of our manuscript, and we appreciate your guidance in improving the quality of our work.

Bibliography rises a question: what sort of references have been used. It is not MLA, Chicago, APA, Turabian. E.g., Pappas IO, Woodside AG. Should have been Pappas, I.O., Woodside, A.G.

Response: Thank you for bringing the issue with the bibliography to our attention. We have thoroughly reviewed and revised the references using the EndNote tool, ensuring adherence to the PLOS ONE guidelines, particularly following the “Vancouver” style for reference formatting. We hope this addresses the concern, and we appreciate your valuable guidance.

Page 5 line 106: …. literature include but (Passive voice is necessary) not limited to local economy

Response: Thank you for your beneficial suggestion. We’ve implemented the recommended changes, incorporating the passive voice as suggested. The revised passage now states, “The influential factors of regional tourism economic disparity, as reported in the existing literature, include, but are not limited to, the local economy, tourism attraction, transportation convenience, policy support, and geographic location” (Page 5, Line 97-99). Your guidance is greatly appreciated.

Line 248: comprehended with set theory[45, 46] (a blank space is necessary)

Response: Thank you for your meticulous attention to detail. We have implemented the necessary correction, and the revised text now reads, “comprehended using set theory [47, 48]” (Page 11, Line 223). Your correction is greatly appreciated.

Line 257: Calibration is refers to the membership (please check the grammar)

Response: Thank you for your valuable comments. We appreciate your careful review, and based on your suggestion, we have made the necessary adjustments. The revised content now states, “Calibration refers to the membership scores of antecedent and result variables when their values are converted into a fuzzy set” (Page 11, Line 230-232). Furthermore, to ensure linguistic precision throughout the manuscript, we enlisted the assistance of Editage, a professional editing service, for a thorough language edit. We trust these enhancements align with your expectations, and we highly value your feedback.

Line 313-315 But different regions have (we suggest adding here “different”) economic foundation, tourism attraction, and geographic location, consequently, ….

Response: Thank you for your helpful suggestion. We appreciate the clarity it brings to our manuscript. Following your advice, we have made the necessary correction. The revised passage now reads, “However, regions have distinct economic foundations, tourist attractions, and geographical locations. Consequently, …” (Page 14, Line 281-282). Your guidance has significantly improved the precision of our expression, and we are grateful for your valuable feedback.

Fig 1, 2 and others should include a period, e.g., Fig. 1, Fig. 2 (check through the entire text)

Response: Thank you for your meticulous feedback. We have carefully reviewed all figure citations throughout the entire text, ensuring full compliance with the PLOS ONE requirements. Your detailed guidance is greatly appreciated.

Line 481 I would rather be careful declaring “The biggest contribution of this research lies in revealing the complex causality….” because the authors’ conclusion does not obligatorily mean the reviewers or large audience’s opinion or conclusion.

Response: Thank you for your meticulous review and valuable insights. We appreciate your caution regarding the statement. In response, we have refined the sentence to express a more measured perspective: “This study makes a significant contribution by shedding light on the complex causality of regional tourism economic disparities through a new perspective on the relationship between these disparities and their influencing factors” (Page 22, Line 435-437). Your thoughtful comment has greatly contributed to the precision of our manuscript.

We extend our sincere gratitude once more to the reviewing expert for the invaluable feedback provided on our manuscript. Your meticulous review and insightful suggestions have not only guided us in addressing specific points but have also significantly contributed to the overall enhancement of our work. We genuinely appreciate the time and effort you dedicated to thoroughly evaluate our research.

Reviewer #3: The originality of this paper is the utilizing statistical data from the tourism economy in 31 provinces in mainland China, this research explores regional tourism economic disparity through asymmetric modeling of fuzzy set qualitative comparative analysis (fsQCA).

The theorectical contributions is the exploration of the causes of regional tourism economic disparity are asymmetric; high and low levels of tourism economy have different formation approaches; the functioning of the most influential factors is impacted by other antecedent conditions, presenting an asymmetric non-linear effect; and favorable transportation convenience is a necessary condition for a high level of tourism economy, while poor tourism attraction is a necessary condition for a low level of tourism economy.

Managerial implications of this research introduces a new perspective on regional tourism economic disparity, enabling in-depth comprehension of the complex interaction and non-linear functional mechanisms of influential factors. Besides, the authors explored a region-based tourism development model. Therefore, it can offer significant practical guidance for tourism policymakers.

Response: Thank you, reviewing expert, for your positive evaluation and recognition of our paper. Your affirmation is truly uplifting and serves as a strong motivation for our ongoing research endeavors. Although specific modification suggestions were not provided this time, we have incorporated modifications based on the suggestions of editors and the other two experts. We extend our sincere gratitude for your time, thoughtful review, and valuable comments. Your support is invaluable to us, and we eagerly anticipate receiving your guidance in our future research endeavors.

Once again, we would like to express our deepest gratitude to the reviewing experts and editors for providing invaluable insights and meticulous feedback. Your expertise has undoubtedly guaranteed the quality of our manuscript. With optimism, we look forward to the favorable consideration of our revised manuscript and hope for its successful acceptance and publication in the PLOS ONE. Thank you very much for your time, consideration, and guidance. Wishing you all the very best.

---

## [Decision Letter · Decision Letter 1]

12 Feb 2024

Asymmetric modeling of regional tourism economic disparity in China

PONE-D-24-00500R1

Dear Dr. Ling,

We’re pleased to inform you that your manuscript has been judged scientifically suitable for publication and will be formally accepted for publication once it meets all outstanding technical requirements.

Kind regards,

Saliha Karadayi-Usta, PhD

Academic Editor

PLOS ONE

Additional Editor Comments (optional):

Reviewers' comments:

Reviewer's Responses to Questions

**Comments to the Author**

1. If the authors have adequately addressed your comments raised in a previous round of review and you feel that this manuscript is now acceptable for publication, you may indicate that here to bypass the “Comments to the Author” section, enter your conflict of interest statement in the “Confidential to Editor” section, and submit your "Accept" recommendation.

Reviewer #1: All comments have been addressed

Reviewer #2: All comments have been addressed

2. Is the manuscript technically sound, and do the data support the conclusions?

Reviewer #1: Yes

Reviewer #2: Yes

3. Has the statistical analysis been performed appropriately and rigorously? 

Reviewer #1: Yes

Reviewer #2: Yes

4. Have the authors made all data underlying the findings in their manuscript fully available?

Reviewer #1: Yes

Reviewer #2: Yes

5. Is the manuscript presented in an intelligible fashion and written in standard English?

Reviewer #1: Yes

Reviewer #2: Yes

6. Review Comments to the Author

Reviewer #1: I accept this paper to publish. Authors added the suggested citations, corrected the English. Paper is suitable for publication.

Reviewer #2: (No Response)

7. PLOS authors have the option to publish the peer review history of their article (what does this mean?). If published, this will include your full peer review and any attached files.

Reviewer #1: No

Reviewer #2: **Yes: **Dr. Kamo Chilingaryan

---

## [Editor Report · Acceptance letter]

27 Feb 2024

PONE-D-24-00500R1 

PLOS ONE

Dear Dr. Ling, 

I'm pleased to inform you that your manuscript has been deemed suitable for publication in PLOS ONE. Congratulations! Your manuscript is now being handed over to our production team.

Kind regards, 

on behalf of

Assoc. Prof. Dr. Saliha Karadayi-Usta 

Academic Editor

PLOS ONE